# REWARD ESTIMATION VIA STATE PREDICTION

## ABSTRACT

Reinforcement learning typically requires carefully designed reward functions in order to learn the desired behavior. We present a novel reward estimation method that is based on a finite sample of optimal state trajectories from expert demonstrations and can be used for guiding an agent to mimic the expert behavior. The optimal state trajectories are used to learn a generative or predictive model of the "good" states distribution. The reward signal is computed by a function of the difference between the actual next state acquired by the agent and the predicted next state given by the learned generative or predictive model. With this inferred reward function, we perform standard reinforcement learning in the inner loop to guide the agent to learn the given task. Experimental evaluations across a range of tasks demonstrate that the proposed method produces superior performance compared to standard reinforcement learning with both complete or sparse hand engineered rewards. Furthermore, we show that our method successfully enables an agent to learn good actions directly from expert player video of games such as the Super Mario Bros and Flappy Bird.

## 1    INTRODUCTION

Reinforcement learning (RL) deals with learning the desired behavior of an agent to accomplish a given task. Typically, a scalar reward signal is used to guide the agent's behavior and the agent learns a control policy that maximizes the cumulative reward over a trajectory, based on observations. This type of learning is referred to as *"model-free"* RL since the agent does not know apriori or learn the dynamics of the environment. Although the ideas of RL have been around for a long time (Sutton & Barto (1998)), great achievements were obtained recently by successfully incorporating deep models into them with the recent success of deep reinforcement learning. Some notable breakthroughs amongst many recent works are, the work from Mnih et al. (2015) who approximated a Q-value function using as a deep neural network and trained agents to play Atari games with discrete control; Lillicrap et al. (2016) who successfully applied deep RL for continuous control agents achieving state of the art; and Schulman et al. (2015) who formulated a method for optimizing control policies with guaranteed monotonic improvement.

In most RL methods, it is very critical to choose a well-designed reward function to successfully learn a good action policy for performing the task. However, there are cases where the reward function required for RL algorithms is not well-defined or is not available. Even for a task for which a reward function initially seems to be easily defined, it is often the case that painful hand-tuning of the reward function has to be done to make the agent converge on an optimal behavior. This problem of RL defeats the benefits of automated learning. In contrast, humans often can imitate instructor's behaviors, at least to some extent, when accomplishing a certain task in the real world, and can guess what actions or states are good for the eventual accomplishment, without being provided with the detailed reward at each step. For example, children can learn how to write letters by imitating demonstrations provided by their teachers or other adults (experts). Taking inspiration from such scenarios, various methods collectively referred to as imitation learning or learning from experts' demonstrations have been proposed (Schaal (1997)) as a relevant technical branch of RL. Using these methods, expert demonstrations can be given as input to the learning algorithm. Inverse reinforcement learning (Ng & Russell (2000); Abbeel & Ng (2004); Wulfmeier et al. (2015)), behavior cloning (Pomerleau (1991)), imitation learning (Ho & Ermon (2016); Duan et al. (2017)), and curiosity-based exploration (Pathak et al. (2017)) are examples of research in this direction.

While most of the prior work using expert demonstrations assumes that the demonstration trajectories contain both the state and action information ($\tau = \{(s_0^i, a_0^i), (s_1^i, a_1^i), ..., (s_t^i, a_t^i)\}$) to solve the imitation learning problem, we, however, believe that there are many cases among real world environments where action information is not readily available. For example, a human teacher cannot tell the student what amount of force to put on each of the fingers when writing a letter.

As such, in this work, we propose a reward estimation method that can estimate the underlying reward based only on the expert demonstrations of state trajectories for accomplishing a given task. The estimated reward function can be used in RL algorithms in order to learn a suitable policy for the task. The proposed method has the advantage of training agents based only on visual observations of experts performing the task. For this purpose, it uses a model of the distribution of the expert state trajectories and defines the reward function in a way that it penalizes the agent's behavior for actions that cause it to deviate from the modeled distribution. We present two methods with this motivation; a generative model and a temporal sequence prediction model. The latter defines the reward function by the similarity between the state predicted by the temporal sequence model trained based on the expert's demonstrations and the currently observed state. We present experimental results of the methods on multiple environments and with varied settings of input and output. The primary contribution of this paper is in the estimation of the reward function based on state similarity to expert demonstrations, that can be measured even from raw video input.

## 2 RELATED WORK

Model-free Reinforcement Learning (RL) methods learn a policy $\pi(a_t|s_t)$ that produces an action from the current observation. Mnih et al. (2015) showed that a q-value function $q(s_t, a_t)$ can be approximated with a deep neural network, which is trained using hand-engineered scalar reward signals given to the agent based on its behavior. Similarly, actor-critic networks in Deep Deterministic Policy Gradients (DDPG) can enable state of the art continuous control, e.g. in robotic manipulation by minimizing the distance between the end effector and the target position. Since the success with DDPG, other methods such as Trust Region Policy Optimization (TRPO) (Schulman et al. (2015)) and Proximal Policy Optimization (PPO) (Schulman et al. (2017)) have been proposed as further improvements for model-free RL in continuous control problems.

Although RL enables an agent to learn an optimal policy without supervised training data, in the standard case, it requires a difficult task of hand-tuning good reward functions for each environment. This has been pointed out previously in the literature (Abbeel & Ng (2004)). Several kinds of approaches have been proposed to workaround or tackle this problem. An approach that does not require reward hand-tuning is behavior cloning based on supervised learning instead of RL. It learns the conditional distribution of actions given states in a supervised manner. Although it has an advantage of fast convergence (Duan et al. (2017)) (as behavior cloning learns a single action from states in each step), it typically results in compounding of errors in the future states.

An alternate approach is Inverse Reinforcement Learning (IRL) proposed in the seminal work by Ng & Russell (2000). In this work, the authors try to recover the optimal reward function as a best description behind the given expert demonstrations from humans or other expert agents, using linear programming methods. It is based on the assumption that expert demonstrations are solutions to a Markov Decision Process (MDP) defined with a hidden reward function (Ng & Russell (2000)). It demonstrated successful estimation of the reward function in case of relatively simple environments such as the grid world and the mountain car problem. Another use of the expert demonstrations is initializing the value function; this was described by Wiewiora (2003). Extending the work by Ng & Russell (2000), entropy-based methods that compute the suitable reward function by maximizing the entropy of the expert demonstrations have been proposed by Ziebart et al. (2008). In the work by Abbeel & Ng (2004), a method was proposed for recovering the cost function based on expected feature matching between observed policy and the agent behavior. Furthermore, they showed this to be the necessary and sufficient condition for the agent to imitate the expert behavior. More recently, there was some work that extended this framework using deep neural networks as non-linear function approximator for both policy and the reward functions (Wulfmeier et al. (2015)). In other relevant work by Ho & Ermon (2016), the imitation learning problem was formulated as a two-players competitive game where a discriminator network tries to distinguish between expert trajectories and agent-generated trajectories. The discriminator is used as a surrogate cost function

which guides the agent's behavior in each step to imitate the expert behavior by updating policy parameters based on Trust Region Policy Optimization (TRPO) (Schulman et al. (2015)). Related recent works also include model-based imitation learning (Baram et al. (2017)) and robust imitation learning (Wang et al. (2017)) using generative adversarial networks. All the above-mentioned methods, however, rely on both state and action information provided by expert demonstrations. Contrarily, we learn only from expert state trajectories in this work.

A recent line of work aims at learning useful policies for agents even in the absence of expert demonstrations. In this regard, Pathak et al. (2017) trained an agent with a combination of reward inferred with intrinsic curiosity and a hand-engineered, complete or even very sparse scalar reward signal. The curiosity-based reward is designed to have a high value when the agent encounters unseen states and a low value when it is in a state similar to the previously explored states. The work reported successful navigation in games like Mario and Doom without any expert demonstrations. In this paper, we compare our proposed methods with the curiosity-based approach and show the advantage over it in terms of the learned behaviors. However, our methods assumed state demonstrations are available as expert data while the curiosity-based method did not use any demonstration data.

## 3 REWARD ESTIMATION METHOD

### 3.1 PROBLEM STATEMENT

We consider an incomplete Markov Decision Process (MDP), consisting of states $\mathcal{S}$ and action space $\mathcal{A}$, where the reward signal $r : \mathcal{S} \times \mathcal{A} \to \mathbb{R}$, is unknown. An agent can act in an environment defined by this MDP following a policy $\pi(\boldsymbol{a_t}|\boldsymbol{s_t})$. Here, we assume that we have knowledge of a finite set of optimal or expert state trajectories $\tau = \{\boldsymbol{S^0}, \boldsymbol{S^1}, ..., \boldsymbol{S^n}\}$. Where, $S^i = \{\boldsymbol{s_0^i}, \boldsymbol{s_1^i}, ..., \boldsymbol{s_m^i}\}$, with $i \in \{1, 2, .., n\}$. These trajectories can represent joints angles, raw images or any other information depicting the state of the environment.

Since the reward signal is unknown, our primary goal is to find a reward signal that enables the agent to learn a policy $\pi$, that can maximize the likelihood of these set of expert trajectories $\tau$. In this paper, we assume that the reward signal can be inferred entirely based on the current state and next state information, $r : \mathcal{S} \times \mathcal{S} \to \mathbb{R}$. More formally, we would like to find a reward function that maximizes the following objective:

$$r^* = \arg\max_r \mathbb{E}_{p(\boldsymbol{s_{t+1}}|\boldsymbol{s_t})} r(\boldsymbol{s_{t+1}}|\boldsymbol{s_t}) \tag{1}$$

where $r(\boldsymbol{s_{t+1}}|\boldsymbol{s_t})$ is the reward function of the next state given the current state and $p(\boldsymbol{s_{t+1}}|\boldsymbol{s_t})$ is the transition probability. We assume the performing to maximize the likelihood of next step prediction in equation 1 will be leading the maximizing the future reward when the task is deterministic. Because this likelihood is based on similarity with demonstrations which are obtained while an expert agent is performing by maximizing the future reward. Therefore we assume the agent will be maximizing future reward when it takes the action that gets the similar next step to expert demonstration trajectory data $\tau$.

### 3.2 PROPOSED METHODS

Let, $\tau = \{\boldsymbol{s_t^i}\}_{i=1:M, t=1:N}$ be the optimal states visited by the expert agent, where $M$ is the number of demonstration episodes, and $N$ is the number of steps within each episode. We estimate an appropriate reward signal based on the expert state trajectories $\tau$, which in turn is used to guide a reinforcement learning algorithm and learn a suitable policy.

We evaluate two approaches to implement this idea. A straightforward approach is to first train a generative model using the expert trajectories $\tau$. Rewards can then be estimated based on similarity measures between a reconstructed state value and the actual currently experienced state value of the agent. This method constrains exploration to the states that have been demonstrated by an expert and enables learning a policy that closely matches the expert. However, in this approach, the temporal order of states are ignored or not readily accounted for. This temporal order of the next state in the sequence is important for estimating the state transition probability function. As such, the next approach we take is to consider a temporal sequence prediction model that can be trained to predict

the next state value given current state, based on the expert trajectories. Once again the reward value can be estimated as a function of the similarity measure between the predicted next state and the one actually visited by the agent. The following sub-sections describes both these approaches in detail.

### 3.2.1 GENERATIVE MODELING

We train a deep generative model (three-layered fully connected auto-encoder) using the state values $s_t^i$ for each step number $t$, sampled from the expert agent trajectories $\tau$. The generative model is trained to minimize the following reconstruction loss (maximize the likelihood of the training data):

$$\boldsymbol{\theta_g^*} = \arg\min_{\boldsymbol{\theta_g}} \Big[ - \sum_{j=1}^{M} \sum_{t=1}^{N} \log p(\boldsymbol{s_t^i}; \boldsymbol{\theta_g}) \Big], \tag{2}$$

where $\boldsymbol{\theta_g^*}$ represents the optimum parameters of the generative model. Following typical settings, we assume $p(\boldsymbol{s_t^i}; \boldsymbol{\theta_g})$ to be a Gaussian distribution, such that equation (2) leads to minimizing the mean square error, $\|\boldsymbol{s_t^i} - g(\boldsymbol{s_t^i}; \boldsymbol{\theta_g})\|_2$, between the actual state $\boldsymbol{s_t^i}$ and the generated state $g(\boldsymbol{s_t^i}; \boldsymbol{\theta_g})$.

The reward value is estimated as a function of the difference between the actual state value $s_{t+1}$ and the generated output $g(\boldsymbol{s_t^i}; \boldsymbol{\theta_g})$,

$$r_t^g = \psi\Big( - \|\boldsymbol{s_{t+1}} - g(\boldsymbol{s_{t+1}}; \boldsymbol{\theta_g})\|_2 \Big), \tag{3}$$

where $\boldsymbol{s_t}$ is the current state value, and $\psi$ can be a linear or nonlinear function, typically hyperbolic tangent or gaussian function. In this formulation, if the current state is similar to the reconstructed state value, i.e. $g(\boldsymbol{s_t}; \boldsymbol{\theta_g})$, the estimated reward value will be higher. However, if the current state is not similar to the generated state, the reward value will be estimated to be low. Moreover, as a reward value is estimated at each time step, this approach can be used even in problems which originally had a highly sparse engineered reward structure.

### 3.2.2 TEMPORAL SEQUENCE PREDICTION

In this approach, we learn a temporal sequence prediction model (the specific networks used are mentioned in the corresponding experiments sections) such that we can maximize the likelihood of the next state given the current state. As such the network is trained using the following objective function,

$$\boldsymbol{\theta_h^*} = \arg\min_{\boldsymbol{\theta_h}} \Big[ - \sum_{j=1}^{M} \sum_{t=1}^{N} \log p(\boldsymbol{s_{t+1}^i} | \boldsymbol{s_t^i}; \boldsymbol{\theta_h}) \Big], \tag{4}$$

where $\boldsymbol{\theta_h^*}$ represents the optimal parameters of the prediction model. We also assume the probability of the next state given the previous state value, $p(\boldsymbol{s_{t+1}^i} | \boldsymbol{s_t^i}; \boldsymbol{\theta_h})$ to be a Gaussian distribution. As such the objective function can be seen to be minimizing the mean square error, $\|\boldsymbol{s_{t+1}^i} - h(\boldsymbol{s_t^i}; \boldsymbol{\theta_h})\|_2$, between the actual next state $\boldsymbol{s_{t+1}^i}$, and the predicted next state $h(\boldsymbol{s_t^i}; \boldsymbol{\theta_h})$.

Thus, the reward function is,

$$r_t^h = \psi\Big( - \|\boldsymbol{s_{t+1}} - h(\boldsymbol{s_t}; \boldsymbol{\theta_h})\|_2 \Big). \tag{5}$$

The estimated reward here, can also be interpreted akin to the generative model case. Here, if the agent's policy takes an action that changes the environment towards states far away from the expert trajectories, the corresponding estimated reward value is low. If the actions of agent bring it close to the expert demonstrated trajectories, thereby making the predicted next state match with the actual visited state value, the reward is estimated to be high. This process of reward shaping or guidance can enable the agent to learn a policy that is optimized based on the expert demonstration trajectories.

Algorithm 1 explains the step by step flow of the proposed methods.

---

**Algorithm 1** Reinforcement Learning with Reward Estimation via State Prediction

1: **procedure** TRAINING DEMONSTRATIONS
2:     Given *trajectories* $\tau$ from expert agent
3:     **for** $s_t^i, s_{t+1}^i \in \tau$ **do**
4:         $\boldsymbol{\theta}^* \leftarrow \arg\min_{\boldsymbol{\theta_g}} \left[ -\sum_{i,t} \log p(s_t^i; \boldsymbol{\theta_g}) \right]$ *or* $\arg\min_{\boldsymbol{\theta_h}} \left[ -\sum_{i,t} \log p(s_{t+1}^i | s_t^i; \boldsymbol{\theta_h}) \right]$
5:     **end for**
6: **end procedure**
7: **procedure** REINFORCEMENT LEARNING
8:     **for** $t = 1, 2, 3, ...$ **do**
9:         Observe *state* $s_t$
10:        Select/execute *action* $a_t$, and observe *state* $s_{t+1}$
11:        $r_t \leftarrow \psi\Big( -\|s_{t+1} - g(s_{t+1}; \boldsymbol{\theta_g})\|_2 \Big)$ *or* $\psi\Big( -\|s_{t+1} - h(s_t; \boldsymbol{\theta_h})\|_2 \Big)$
12:        Update the deep reinforcement learning network using the tuple $(s_t, a_t, r_t, s_{t+1})$
13:    **end for**
14: **end procedure**

---

## 4 EXPERIMENTS

In order to evaluate our reward estimation methods, we conducted experiments across a range of environments. We consider five different tasks, namely: robot arm reaching task (reacher) to a fixed target position, robot arm reaching task to a random target position, controlling a point agent for reaching a target while avoiding an obstacle, learning an agent for longest duration of flight in the Flappy Bird video game, and learning an agent for maximizing the traveling distance in Super Mario Bros video game. Table 1 summarizes the primary differences between the five experimental settings.

| Environment | Input | Action | RL method |
|---|---|---|---|
| Reacher (fixed target) | Joint angles & distance to target | Continuous | DDPG |
| Reacher (random target) | Joint angles | Continuous | DDPG |
| Mover with avoiding obstacle | Positon & distance to target | Continuous | DDPG |
| Flappy Bird | Image & bird position | Discrete | DQN |
| Super Mario Bros. | Image | Discrete | A3C |

Table 1: Table comparing the different enviornments

### 4.1 REACHER

We consider a 2-DoF robot arm in the x-y plane that has to learn to reach with the end-effector a point target. The first arm of the robot is a rigidly linked $(0, 0)$ point, with the second arm linked to its edge. It has two joint values $\boldsymbol{\theta} = (\theta_1, \theta_2)$, $\theta_1 \in (-\infty, +\infty), \theta_2 \in [-\pi, +\pi]$ and the lengths of arms are 0.1 and 0.11 units, respectively. The robot arm was initialized by random joint values at the initial step for each episode. In the following experiments, we have two settings: fixed point target, and a random target. The applied continuous action values $a_t$ is used to control the joint angles, such that, $\dot{\boldsymbol{\theta}} = \boldsymbol{\theta_t} - \boldsymbol{\theta_{t-1}} = 0.05\,a_t$. Each action value has been clipped the range $[-1, 1]$. The reacher task is enabled using the physics engine within the roboschool environment (Brockman et al. (2016); OpenAI (2017)). Figure 1 describes the roboshool environment. The robot arms are in blue, the blue-green point is the end-effector, and the pink dot is the desired target location.

### 4.1.1 REACHER TO FIXED POINT

In this experiment, the target point $p_{tgt}$ is always fixed at $(0.1, 0.1)$. The state vector $s_t$ consists of the following values: absolute end position of first arm $(p_2)$, joint value of elbow $(\theta_2)$, velocities of the joints $(\dot{\theta}_1, \dot{\theta}_2)$, absolute target position $(p_{tgt})$, and the relative end-effector position from target $(p_{ee} - p_{tgt})$. We used DDPG (Lillicrap et al. (2016)) for this task, with the number of steps for each episode being 500 in this experiment [1]. The reward functions used in this task were as follows:

$$\text{Dense reward} : r_t = -\|p_{ee} - p_{tgt}\|_2 + r_t^{env}, \tag{6}$$

$$\text{Sparse reward} : r_t = -\tanh(\alpha\|p_{ee} - p_{tgt}\|_2) + r_t^{env}, \tag{7}$$

$$\text{Generative Model (GM)} : r_t = -\tanh(\beta\|s_{t+1} - g(s_{t+1}; \theta_{2k})\|_2), \tag{8}$$

$$\text{GM with } r_t^{env} : r_t = -\tanh(\beta\|s_{t+1} - g(s_{t+1}; \theta_{2k})\|_2) + r_t^{env}, \tag{9}$$

$$\text{GM (1000 episodes)} : r_t = -\tanh(\beta\|s_{t+1} - g(s_{t+1}; \theta_{1k})\|_2) + r_t^{env}, \tag{10}$$

$$\text{GM with action} : r_t = -\tanh(\beta\|[s_{t+1}, a_t] - g([s_{t+1}, a_t]; \theta_{2k,+a})\|_2) + r_t^{env}, \tag{11}$$

where $r_t^{env}$ is the environment specific reward, which is calculated based on the cost for current action, $-\|a_t\|_2$. This regularization is required for finding the shortest path to reach the target. As this cost is critical for fast convergence, we use this in all cases. The dense reward is a distance between end-effector and the target, and the sparse reward is based on a bonus for reaching. The generative model parameters $\theta_{2k}$ is trained by $\tau^{2k}$ trajectories that contains only states of 2000 episodes from an agent trained during 1k episodes with dense reward. The generative network has $400, 300$ and $400$ units fully-connected layers, respectively. They also have ReLU activation function, with the batch size being 16, and number of epochs being 50. $\theta_{1k}$ is trained from $\tau^{1k}$ trajectories that is randomly picked 1000 episodes from $\tau^{2k}$. The GM with action uses a generative model $\theta_{2k,+a}$ that is trained pairs of state and action for 2000 episodes for same agents as $\tau^{2k}$. We use a tanh nonlinear function for the estimated reward in order to keep a bounded value. The $\alpha, \beta$ change sensitiveness of distance or reward, were both set to $100$ [2]. Here, we also compare our results with behavior cloning (BC) method Pomerleau (1991) where the trained actor networks directly use obtained pairs of states and actions.

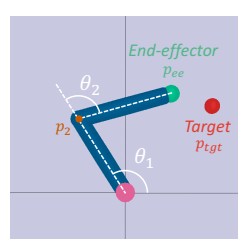

Figure 1: The environment of reacher task. The reacher has two arms, and objective of agent is reaching end-effector *(green)* to target point *(red)*.

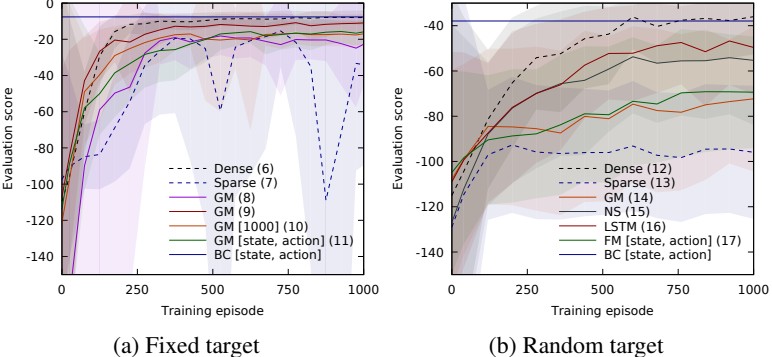

(a) Fixed target  (b) Random target

Figure 2: Performance of RL for reacher. The dash lines are results using the *human* designed reward, solid lines are results using the estimated reward based on demonstration data. The evaluation scores *(y-axis)* are normalized based on max and min reward. The corresponding equation numbers are referred to within bracket.

Figure 2a shows the difference in performance by using the different reward functions [3]. All methods are evaluated by a score *(y-axis)*, which was normalized to a minimum (0) and a maximum (1) value. The proposed method, especially "GM with $r_t^{env}$", manages to achieve a score nearing that of the dense reward, with the performance being much better as compared to the sparse reward setting.

---

[1]Network details are in appendix (see 6.1).

[2]We tried $\{1, 10, 100\}$, and then selected the best for each

[3]BC doesn't have an update of actor, hence it is straight line.

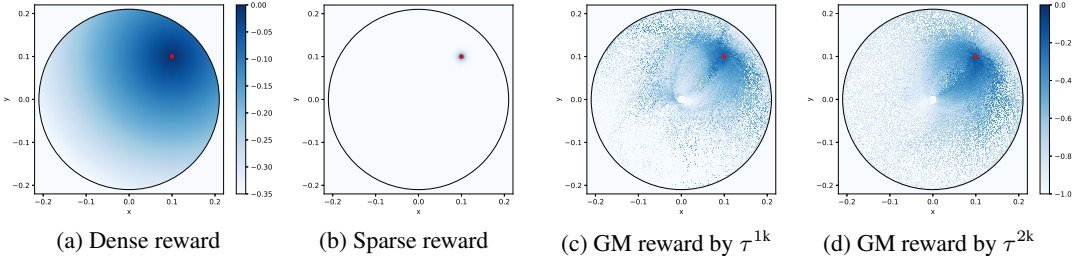

|                     |                      |                                    |                                    |
|---------------------|----------------------|------------------------------------|------------------------------------|
| (a) Dense reward    | (b) Sparse reward    | (c) GM reward by $\tau^{1k}$       | (d) GM reward by $\tau^{2k}$       |

Figure 3: These show the reward values *(blue)* for each end-effector position, and target position *(red)*. The GM rewards are dependent on state value $s_{t+1}$. Therefore, these are average values taken over 1000 different states values for the same end-effector position. Color bars for panels (b), (c) and (d) are the same.

Moreover, the learning curves based on the rewards estimated with the generative model show a faster convergence rate.

However, the result without environment specific reward, i.e. with the additional action regularization term, takes a longer time to converge. This is primarily because of the fact that GM reward is reflective of the distance between target and end-effector, and cannot directly account for the action regularization. The GM reward based on $\tau^{1k}$ underperforms as compared with GM reward based on $\tau^{2k}$ because of the lack of demonstration data. Figure 3 shows the reward value of each end-effector point. GM estimated reward using $\tau^{2k}$ has better reward map as compared to GM estimated reward using $\tau^{1k}$. However, these demonstrations data (Figure 8) are biased by the robot trajectories. Thus, a method of generating demonstration data that normalizes or avoids such bias will further improve the reward structure.

If the demonstrations contain the action information in addition to state information, behavior cloning achieves good performance. Surprisingly, however, when using both state and action information in the generative model, "GM [state, action]", the performance of the agent is comparatively poor.

### 4.1.2 REACHER TO RANDOM POINT

In this experiment, the target point $p_{tgt}$ is initialized by a random uniform distribution of $[-0.27, +0.27]$, that includes points outside of the reaching range of the robot arms. Furthermore, we removed the relative position of the target from the input state information. This makes the task more difficult. The state vector $s_t$ has the following values: $p_2, \theta_2, \dot{\theta}_1, \dot{\theta}_2, p_{tgt}$. In the previous experiment, the distribution of states in expert trajectories is expected to be similar to the reward structure due to a fixed target location. However, when the target position changes randomly, this distribution is not fixed. We, therefore, evaluate with the temporal sequence prediction model $h(s_t; \theta_h)$ in this experiment. The RL setting is same as the previous experiment, however we changed the total number of steps within each episode to $400$. The reward functions used in this experiment were calculated as follows:

$$\text{Dense reward}: r_t = -\|p_{ee} - p_{tgt}\|_2 + r_t^{env}, \tag{12}$$

$$\text{Sparse reward}: r_t = \tanh(-\alpha\|p_{ee} - p_{tgt}\|_2) + r_t^{env}, \tag{13}$$

$$\text{GM reward}: r_t = \tanh(-\beta\|s_{t+1} - g(s_{t+1}; \theta_g)\|_2) + r_t^{env}, \tag{14}$$

$$\text{Next State (NS) reward}: r_t = \tanh(-\gamma\|s_{t+1} - h(s_t; \theta_h)\|_2) + r_t^{env}, \tag{15}$$

$$\text{LSTM reward}: r_t = \tanh(-\gamma\|s_{t+1} - h(s_{t:t-n}; \theta_{lstm})\|_2) + r_t^{env}, \tag{16}$$

$$\text{Forward Model (FM) reward}: r_t = \tanh(-\gamma\|s_{t+1} - f(s_t, a_t; \theta_{+a})\|_2) + r_t^{env}. \tag{17}$$

The expert demonstrations $\tau$ were obtained using the states of 2000 episodes running a trained agent with dense hand-engineered reward. The GM estimated reward uses the same setting as in the previous experiment. NS is a model that predicts the next state given current state, and was trained using demonstration data $\tau$ [4]. The LSTM model uses Long short-term memory (Hochreiter

---

[4]The hidden layers are same as GM model.

& Schmidhuber (1997)) as a temporal sequence prediction model. The state in reacher task does not contain time sequence data, hence we use a finite state history as input. LSTM model has three layers [5] and one fully- connected layer with 40 ReLU activation units. The forward model based reward estimation is based on predicting the next state given both the current state and action [6]. Here, we also compared with the baseline behavior cloning method. In this experiment, $\alpha$ is 100, $\beta$ is 1, and $\gamma$ is 10.

Figure 2b shows the performance of the trained agent using the different reward functions. In all cases using estimated rewards performances significantly better than the sparse reward case. The LSTM based prediction method gives the best results, reaching close to the performance obtained with dense hand engineered reward function. As expected, the GM based reward estimation fails to work well in this relatively complex experimental setting. The NS model estimated reward, which predicts next state given only the current state information, has comparable performance with LSTM based prediction model during the initial episodes. The FM based reward function also performs poorly in this experiment. Comparatively, the direct BC works relatively well. This indicates that it is better to use behavior cloning than reward prediction when both state and action information are available from demonstration data.

## 4.2 REACHER WITH OBSTACLE

Starting with this experiment, we evaluate using only the temporal sequence prediction method. As such, here we use a finite history of the state values in order to predict the next state value. We assume that predicting a part of the state that is related to a given action allows the model to make a better estimate of the reward function. Former work by Pathak et al. (2017) predicts a function of the next state, $\phi(s_{t+1})$ rather than predicting the raw value $s_{t+1}$, as in this paper. In this experiment, we also changed the non-linear function $\psi$ in the proposed prediction method to a Gaussian function (as compared to the hyperbolic tangent function used in previous experiments). This allows us to compare the robustness of our proposed method for reward estimation to different non-linear functions.

We develop a new environment that adds an obstacle to the reaching task. This reacher is a two-dimensional point $(x, y)$ that uses position control. In Figure 4 we show the modified environment setup. The agent's goal is to reach the target while avoiding the obstacle in this case. The initial position of agent, the target position, and an obstacle position were initialized randomly. The state value contains the agent absolute position $(p_t)$, current velocity of the agent $(\dot{p}_t)$, a target absolute position $(p_{tgt})$, an obstacle absolute position $(p_{obs})$, and the relative location of target and obstacle with respect to the agent $(p_t - p_{tgt}, p_t - p_{obs})$. Once again the RL algorithm used in this experiment was DDPG (Lillicrap et al. (2016)) for continuous control [7]. The number of steps for each episode set to 500. Here, we used the following reward functions:

$$\text{Dense reward} : r_t = -\|p_t - p_{tgt}\|_2 + \|p_t - p_{obs}\|_2, \tag{18}$$

$$\text{LSTM reward} : r_t = \exp(-\|s_{t+1} - h(s_{t:t\text{-}n}; \theta_{lstm})\|_2 / 2\sigma_1^2), \tag{19}$$

$$\text{LSTM (state-selected) reward} : r_t = \exp(-\|s'_{t+1} - h'(s_{t:t\text{-}n}; \theta_{lstm})\|_2 / 2\sigma_2^2), \tag{20}$$

where $h'(s_{t:t\text{-}n}; \theta_{lstm})$ is a network that predicts a selected part of state values given a finite history of state information. The dense reward is composed of both, the target distance cost and the obstacle distance bonus. The optimal state trajectories $\tau$ contains 800 *"human guided"* demonstration data. In this case, the LSTM network consisted of two layers, each with 256 units with ReLU activations. In this experiment, $\sigma_1$ is 0.005, and $\sigma_2$ is 0.002 [8].

Figure 5 shows the performance with the different estimated or hand-engineered reward settings. The LSTM based prediction method learns to reach the target faster than the dense reward, while LSTM $(s')$ has the best overall performance by learning with human-guided demonstration data.

---

[5]Two LSTM layers with 128 units, 30% dropout, and `tanh` activation function.

[6]Hidden layers structure is same as the GM model.

[7]Network details are in appendix (see 6.3 section)

[8]We tried $\{0.002, 0.005\}$, and then selected the best for each.

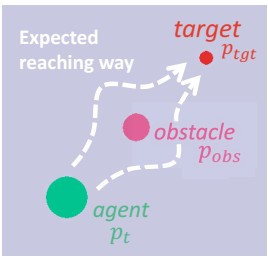

Figure 4: The environment of reacher with an obstacle. The agent *(green)* will move to reach the target *(red)*, while avoiding the obstacle *(pink)*.

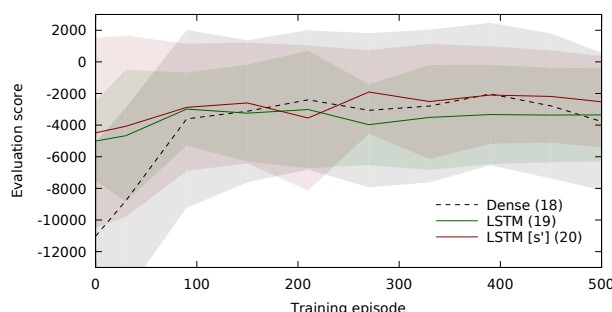

Figure 5: The performance for reacher with obstacle environment.

### 4.3 FLAPPY BIRD

We use a re-implementation (Lau (2017)) of Android game, "Flappy Bird", in python (pygame). The objective of this game is to pass through the maximum number of pipes without collision. The control is a single discrete command of whether to flap the bird wings or not. The state has four consecutive gray frames (4 x 80 x 80). The RL is trained by DQN (Mnih et al. (2015)) [9], and the update frequency of deep network is 100 steps. The used rewards are,

$$\text{Dense reward} : r_t = \begin{cases} +0.1 & \text{if } \textit{alive}; \\ +1 & \text{if } \textit{pass through a pipe}; \\ -1 & \text{if } \textit{collide to a pipe}. \end{cases} \tag{21}$$

$$\text{LSTM reward} : r_t = \exp(-\|\boldsymbol{s'_{t+1}} - h'(\boldsymbol{s_t}; \boldsymbol{\theta_{lstm}})\|_2/2\sigma^2), \tag{22}$$

which $\boldsymbol{s'_{t+1}}$ is an absolute position of the bird, which can be given from simulator or it could be processed by pattern matching or CNN from raw images, $h'(\boldsymbol{s_t}; \boldsymbol{\theta_{lstm}})$ is a predicted the absolute position. Hence, LSTM is trained for predicting absolute position of bird location given images. The $\tau$ of this experiment is 10 episodes data from a trained agent in the repository by Lau (2017). Also, we also compared with the baseline behavior cloning method. In this experiment, $\sigma$ is 0.02.

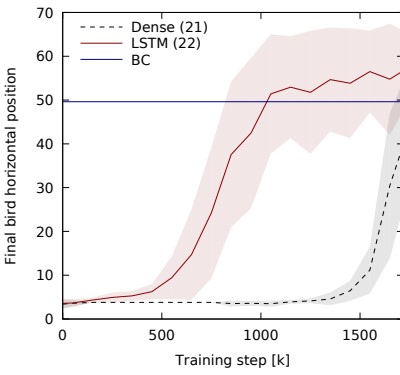

Figure 6: The performance for Flappy Bird.

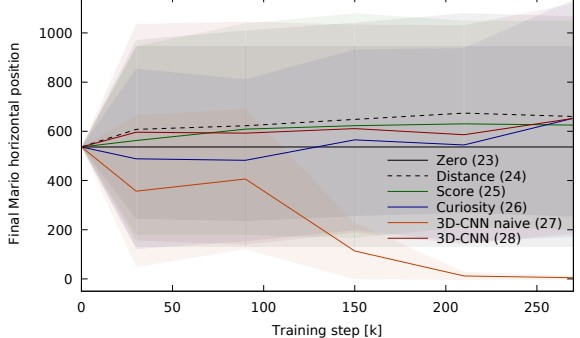

Figure 7: The performance for Super Mario Bros.

Figure 6 shows the result of LSTM reward is better than normal "hand-crafted" reward. The reason for this situation is, the normal dense reward just describes the traveling distance, but our LSTM reward will teach which absolute transition of bird is good. Also, LSTM has better convergence than BC result; the reason is the number of demonstrations is not enough for behavior cloning method.

---

[9]Network details are in appendix (see 6.4 section)

## 4.4 MARIO

In the final task, we consider a more complex environment in order to evaluate our proposed reward estimation method using only state information. Here we use the *Super Mario Bros* classic Nintendo video game environment (Paquette (2017)). Our proposed method estimates reward values based on expert gameplay video data (using only the state information in the form of image frames).

In this experiment, we also benchmarked against the recently proposed curiosity-based method (Pathak et al. (2017)) using the implementation provided by the same authors (Pathak (2017)). This was used as the baseline reinforcement learning technique. Unlike in the actual game, here we always initialize Mario to the starting position rather than a previously saved checkpoint. This is a discrete control setup, where, Mario can make 14 types of actions [10]. The state information consists of sequential input of four 42 x 42 gray image frames [11]. Here we used the A3C RL algorithm (Mnih et al. (2016)). We used gameplay of stage "1-1" for this experiment, with the objective of the agent being to travel as far as possible and achieve as high a score as possible.

The rewards functions used in this experiment were as follows:

$$\text{Zero}: r_t = 0, \tag{23}$$
$$\text{Distance}: r_t = position_t - position_{t-1}, \tag{24}$$
$$\text{Score}: r_t = score_t, \tag{25}$$
$$\text{Curiosity (Pathak et al. (2017))}: r_t = \eta\|\phi(\boldsymbol{s_{t+1}}) - f(\phi(\boldsymbol{s_t}), \boldsymbol{a_t}; \theta_F)\|_2, \tag{26}$$
$$\text{3D-CNN (naïve)}: r_t = 1 - \|\boldsymbol{s_{t+1}} - h(\boldsymbol{s_t}; \theta)\|_2, \tag{27}$$
$$\text{3D-CNN}: r_t = max(0, \zeta - \|\boldsymbol{s_{t+1}} - h(\boldsymbol{s_t}; \theta)\|_2), \tag{28}$$

where $position_t$ is the current position of Mario at time $t$, $score_t$ is the current score value at time $t$, and $\boldsymbol{s_t}$ are screen images from the Mario game at time $t$. The position and score information are obtained using the Mario game emulator. In this experiment, we use a three-dimensional convolutional neural network (Ji et al. (2013)) (3D-CNN) for our temporal sequence prediction method. In order to capture expert demonstration data, we took 15 game playing videos by five different people [12]. In total, the demonstration data consisted of 25000 frames. The length of skipped frames in input to the temporal sequence prediction model was 36, as humans cannot play as fast as an RL agent; however, we did not change the skip frame rate for the RL agent. The 3D-CNN consisted of 4 layers[13] and a final layer to reconstruct the image. The agent was trained using 50 epochs with a batch size of 8. We implemented two prediction methods for reward estimation. In the naïve method the Mario agent will end up getting positive rewards if it sits in a fixed place without moving. This is because it can avoid dying by just not moving. However, this is clearly a trivial suboptimal policy. Hence, we implemented the alternate reward function based on the same temporal sequence prediction model, but we apply a threshold value that prevents the agent from converging onto such a trivial solution. Here, the value of $\zeta$ is 0.025, which was calculated based on the reward value obtained by just staying fixed at the initial position.

Figure 7 shows the performance with the different reward functions. Here, the graphs directly show the average results over multiple trials. As observed, the agent was unable to reach large distances even while using *"hand-crafted"* dense rewards and did not converge to the goal every time [14]; this behavior was also observed by Pathak et al. (2017) for their reward case. As observed from the average curves of Figure 7, the proposed 3D-CNN method learns relatively faster as compared to the curiosity-based agent (Pathak et al. (2017)). As expected the 3D-CNN (naïve) method converged to a solution of remaining fixed at the initial state. As future work, we hope to improve the performance in this game setting using deeper RL networks, as well as large input image sizes. Overall estimating reward from $\phi(\boldsymbol{s_t})$ without the need of action data, allows an agent to learn suitable policy directly from raw video data. The abundance of visual data creates ample opportunity for this type of reward estimation method to be explored further in different video game settings.

---

[10]A single action is repeated for six consecutive frames. Please refer to (Pathak et al. (2017)) for details.

[11]Every next six frames were skipped.

[12]All videos consisted of games where the player succeeded in clearing the stage.

[13]Two layers with (2 x 5 x 5), two layers with (2 x 3 x 3) kernels, all have 32 filters, and every two layers with (2, 1, 1) stride.

[14]By our experiment, even if it trained long steps, such as 3.0M; it just reached around 600 - 700 averagely.

## 5 CONCLUSION

In this paper, we proposed two variations of a reward estimation method via state prediction by using state-only trajectories of the expert; one based on an autoencoder-based generative model and one based on temporal sequence prediction using LSTM. Both the models were for calculating similarities between actual states and predicted states. We compared the methods with conventional reinforcement learning methods in five various environments. As overall trends, we found that the proposed method converged faster than using hand-crafted reward in many cases, especially when the expert trajectories were given by humans, and also that the temporal sequence prediction model had better results than the generative model. It was also shown that the method could be applied to the case where the demonstration was given by videos. However, detailed trends were different for the different environments depending on the complexity of the tasks. Neither model of our proposed method was versatile enough to be applicable to every environment without any changes of the reward definition. As we saw in the necessity of the energy term of the reward for the reacher task and in the necessity of special handling of the initial position of Mario, the proposed method has a room of improvements especially in modeling global temporal characteristics of trajectories. We would like to tackle these problems as future work.

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

# 6 APPENDIX

## 6.1 NETWORK DETAIL FOR REACHER

The DDPG's actor network has $400$ and $300$ unites fully-connected (fc) layers, the critic network has also $400$ and $300$ fully-connected layers, and each layer has a ReLU (Nair & Hinton (2010)) activation function. We put the tanh activation function at the final layer of actor network. Without this modification, the normal RL takes a long time to converge. Also, initial weights will be set from a uniform distribution $U(-0.003, 0.003)$. The exploration policy is Ornstein-Uhlenbeck process (Uhlenbeck & Ornstein (1930)) ($\theta = 0.15, \mu = 0, \sigma = 0.01$), size of reply memory is 1M, and optimizer is Adam (Kingma & Ba (2014)). We implemented these experiments by Keras-rl (Plappert (2016)), Keras (Chollet (2015)), and Tensorflow (Abadi et al. (2015)) libraries.

## 6.2 DEMONSTRATION DATA FOR REACHER TO FIXED TARGET

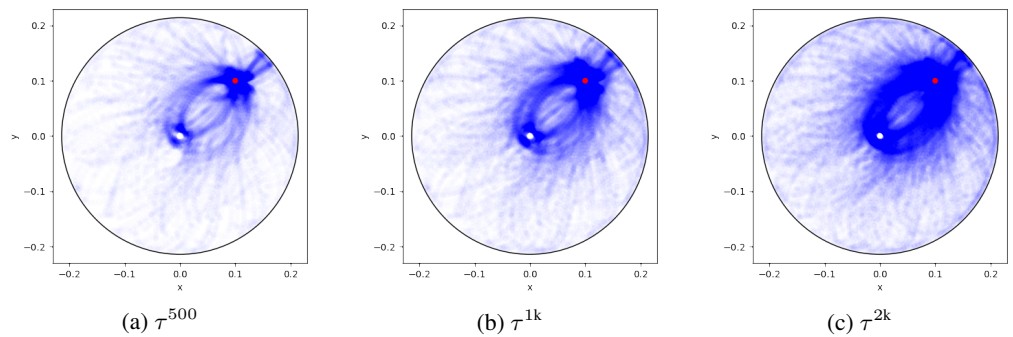

(a) $\tau^{500}$      (b) $\tau^{1k}$      (c) $\tau^{2k}$

Figure 8: These are scatter-plots of end-effector positions *(blue)* for each state of captured demonstration $\tau^{500}, \tau^{1k}, \tau^{2k}$, each point is drawn by $\alpha$ is 0.01. And the fixed target position is also plotted *(red)*. Notes that this is just plotting end-effector position, there is more variation in other state values. For example, even if the end-effector position were same, arms' pose (joint values) might be different. Note that $\tau^{500}$ is not used in the experiment.

## 6.3 NETWORK DETAILS FOR REACHER WITH OBSTACLE

The DDPG's actor network has $64$ and $64$ unites fully-connected layers, the critic network has also $64$ and $64$ fully-connected layers, and each layer has a ReLU activation function. Initial weights will be set from a uniform distribution $U(-0.003, 0.003)$. The exploration policy is Ornstein-Uhlenbeck process (Uhlenbeck & Ornstein (1930)) ($\theta = 0.15, \mu = 0, \sigma = 0.01$), size of reply memory is 500k, and optimizer is Adam.

## 6.4 NETWORK DETAILS FOR FLAPPY BIRD

The DQN has three convolutional (kernel size are 8x8, 4x4, and 3x3, filter num are 32, 64, and 64, and stride num are 4, 2, and 1), one fc layer (512), and final layer. The ReLU activation function is inserted after each layer. It uses the Adam optimizer, and mean square loss. Replay memory size is 2M, batch size is 256, and other parameters have been followed the repository (Lau (2017)).

