# OpenReview forum: "Reward Estimation via State Prediction"
_ICLR.cc/2018/Conference — Reject_

### Official Review · AnonReviewer3 · 2017-11-26
**Nice simulations, but theoretically incomplete.**

**Rating:** 4
**Confidence:** 4

**Review:**

The authors propose to solve the inverse reinforcement learning problem of inferring the reward function from observations of a behaving agent, i.e. trajectories, albeit without observing state-action pairs as is common in IRL but only with the state sequences. This is an interesting problem setting. But, apparently, this is not the problem the authors actually solve, according to eq. 1-5. Particularly eq. 1 is rather peculiar. The main idea of RL in MDPs is that agents do not maximize immediate rewards but instead long term rewards. I am not sure how this greedy action should result in maximizing the total discounted reward along a trajectory.
Equation 3 seems to be a cost function penalizing differences between predicted and observed states. As such, it implements a sort of policy imitation, but that is quite different from the notion of reward in RL and IRL. Similarly, equation 4 penalizes differences between predicted and observed state transitions.
Essentially, the current manuscript does not learn the reward function of an MDP in the RL setting, but it learns some sort of a shaping reward function to do policy imitation, i.e. copy the behavior of the demonstrator as closely as possible. This is not learning the underlying reward function. So, in my view, the manuscript does a nice job at policy fitting, but this is not reward estimation. The manuscript has to be rewritten that way.
One could also argue that the manuscript would profit from a better theoretical analysis of the IRL problem, say:
C. A. Rothkopf, C. Dimitrakakis. Preference elicitation and inverse reinforcement learning. ECML 2011
Overall the manuscript leverages on deep learning’s power of function approximation and the simulation results are nice, but in terms of the soundness of the underlying RL and IRL theory there is some work to do.

---

> ### Author Response · Authors · 2017-12-21
> **answers**
>
> Thank you very much for your comments.
> We are very happy you understood this is nice simulations and interesting problem setting.
> And we put the answers to your questions and suspicions, also we updated the paper by following your comments.
>
> >I am not sure how this greedy action should result in maximizing the total discounted reward along a trajectory.
>
> This is a very important point of this proposed method.
> The expert agent (it will be also human demonstrations in some tasks) will do actions that are maximizing the future reward.
> The proposed method will be trained as similar as possible to the expert agent by equation 1.
> When the agent took similar actions, it will get the high reward.
> And, we elaborated the context by your comments also.
> "We assume the maximizing likelihood of next step prediction in equation 1 will be globally optimized in RL."
> ->
> "We assume the performing to maximize the likelihood of next step prediction in equation 1 will be leading the maximizing the future reward when the task is deterministic. Because this likelihood is based on similarity with demonstrations which are obtained while an expert agent is performing by maximizing the future reward. Therefore we assume the agent will be maximizing future reward when it takes the action that gets the similar next step to expert demonstration trajectory data \tau. "
>
> >Essentially, the current manuscript does not learn the reward function of an MDP in the RL setting, but it learns some sort of a shaping reward function to do policy imitation, i.e. copy the behavior of the demonstrator as closely as possible.
>
> Actually, this is true, we agree this opinion.
> The objective of proposed reward is copying the behavior of the demonstrator.
> However, with our assumption, the agent could not get the "actual" reward during testing, but the expert agent got the actual reward or knew the task.
> Then the reward of the proposed method is based on similarity of behavior with the demonstrator.
> So, the predicted reward likes (hidden) actual reward that is used by the expert agent.
> We used "reward estimation" for such meaning.
>
> And also, if we could use the "actual" reward during testing, the agent can simply combine these rewards and do some explorations for normal RL.

---

### Official Review · AnonReviewer2 · 2017-11-28
**This paper provides an empirically effective method for inverse reinforcement learning when given thousands of expert state trajectories without having access to the expert’s actions, but it is unclear if the method still performs well beyond distribution covered by the expert trajectories.**

**Rating:** 5
**Confidence:** 3

**Review:**

This paper uses inverse reinforcement learning to infer additional shaping rewards from demonstrated expert trajectories.  The key distinction from many previous works in this area is that the expert’s actions are assumed to not be available, and the inferred reward on a transition is assumed to be a function of the previous and subsequent state.  The expert trajectories are first used to train either a generative model or an LSTM on next state prediction. The inferred reward for a newly experienced transition is then defined from the negative error between the predicted and actual next state.  The method is tested on several reacher tasks (low dimensional continuous control), as well as on two video games (Super Mario Bros and Flappy Bird).  The results are positive, though they are often below the performance of behavioral cloning (which only trains from the expert data but also uses the expert’s actions).  The proposed methods perform competitively with hand-designed dense shaping rewards for each task.

The main weakness of the proposed approach is that the addition of extra rewards from the expert trajectories seems to skew the system’s asymptotic behavior away from the objective provided by the actual environment reward.  One way to address this would be to use the expert trajectories to infer not only a reward function, but also an initial state value function (trained on the expert trajectories with the inferred reward).  This initial value function could be added to the learned value function and would not limit asymptotic performance (unlike the addition of inferred rewards as proposed here).  This connection between reward shaping and initial Q values was described by Wiewirora in 2003 (“Potential-based Shaping and Q-Value Initialization are Equivalent”).

I am also uncertain of the robustness of the proposed approach when the learning agent goes beyond the distribution of states provided by the expert (where the inferred reward model has support).  Will the inferred reward function in these situations go towards zero?  Will the inferred reward skew the learning algorithm to a worse policy?  How does one automatically balance the reward scale provided by the environment with the the reward scaling provided by psi, or is this also assumed to be manually crafted for each domain?  These questions make me uncertain of the utility of the proposed method.

---

> ### Author Response · Authors · 2017-12-21
> **answers**
>
> Thank you very much for your comments.
> We are very happy you understood the effectiveness of the proposed method.
> And we put the answers to your questions and suspicions, also we updated the paper by following your comments.
>
> >The main weakness of the proposed approach is that the addition of extra rewards from the expert trajectories seems to skew the system’s asymptotic behavior away from the objective provided by the actual environment reward
>
> Actually, yes, that's true.
> The proposed method will try to get the "actual" environment reward from the demonstrations from the expert agent that is having \pi^*.
> The reward of the proposed method is not perfectly same as such actual reward, of course.
>
> >This connection between reward shaping and initial Q values was described by Wiewirora in 2003
>
> Thank you for suggesting the new reference.
> We added this paper as references.
> "Another use of the expert demonstrations is initializing the value function; this was described by Wiewiora (2003)."
>
> >I am also uncertain of the robustness of the proposed approach when the learning agent goes beyond the distribution of states provided by the expert (where the inferred reward model has support). Will the inferred reward function in these situations go towards zero?
>
> We agree the robustness of the proposed method is very difficult to understand.
> Hence, we tried to apply to many experiments in different environments.
>
> We expected the inferred reward will be zero, when the state will be beyond the distribution of expert states.
> We confirmed these point experimentally.
> Please see the figure 3 and figure 8, fig 3 shows the reward value for each point in reacher task, and fig 8 shows the kind of distribution.
> The reward value at a place that shown low frequent is nearly zero.
> On the other hand, the reward value in the distribution of expert states is high value.
>
> >Will the inferred reward skew the learning algorithm to a worse policy?
>
> The proposed method will not lead to training worse policy.
> Because the proposed reward estimation network has been trained from demonstrations of given expert agent.
> However, of course, if the given agent has a bad policy, it will learn this policy.
> On the other hand, if the inferred reward skew to a worse policy, the RL will not be converged.
> In all experiments of this paper, the proposed method converged good behaviors.
>
> >How does one automatically balance the reward scale provided by the environment with the the reward scaling provided by psi, or is this also assumed to be manually crafted for each domain?
>
> Actually, if we use the tanh or exp function for \phi, the reward shape was similar.
> But \beta in tanh or \sigma in exp is important for RL training.
> If the \beta is too high or too low, the convergence will be slow or the reward will be jerky.
> In this paper, we tried a few values for each domain and picked one of it.
> (we forgot to describe this setting for \sigma, so we added the way to choose this hyper-parameter)

---

### Official Review · AnonReviewer1 · 2017-11-30
**The paper presents a method for speeding RL algorithms by learning a reward function from expert demonstrations. The learned reward function explicitly penalizes deviations from the demonstrations. Experiments in simulated environments show the benefits of this method, but the concept is not original.**

**Rating:** 3
**Confidence:** 4

**Review:**

To speed up RL algorithms, the authors propose a simple method based on utilizing expert demonstrations. The proposed method consists in explicitly learning a prediction function that maps each time-step into a state. This function is learned from expert demonstrations. The cost of visiting a state is then defined as the distance between that state and the predicted state according to the learned function. This reward is then used in standard RL algorithms to learn to stick close to the expert's demonstrations. An on-loop variante of this method consists of learning a function that maps each state into a next state according to the expert, instead of the off-loop function that maps time-steps into states.
While the experiments clearly show the advantage of this method, this is hardly surprising or novel. The concept of encoding the demonstration explicitly in the form of a reward has been around for over a decade. This is the most basic form of teaching by demonstration. Previous works had used other models for generalizing demonstrations (GMMs, GPs, Kernel methods, neural nets etc..). This paper uses a three layered fully connected auto-encoder (which is not that deep of a model, btw) for the same purpose. The idea of using this model as a reward instead of directly cloning the demonstrations is pretty straightforward.

Other comments:
- Most IRL methods would work just fine by defining rewards on states only and ignoring actions all together. If you know the transition function, you can choose actions that lead to highly rewarding states, so you don't need to know the expert's executed actions.
- "We assume that maximizing likelihood of next step prediction in equation 1 will be globally optimized in RL". Could you elaborate more on this assumption? Your model finds rewards based on local state features, where a greedy (one-step planning) policy would reproduce the expert's demonstrations (if the system is deterministic). It does not compare the global performance of the expert to alternative policies (as is typically done in IRL).
- Related to the previous point: a reward function that makes every step of the expert optimal may not be always exist. The expert may choose to go to terrible states with the hope of getting to a highly rewarding state in the future. Therefore, the objective functions set in this paper may not be the right ones, unless your state description contains features related to future states so that you can incorporate future rewards in the current state (like in the reacher task, where a single image contains all the information about the problem). What you need is actually features that can capture the value function (like in DQN) and not just the immediate reward (as is done in IRL methods).
- What if in two different trajectories, the expert chooses opposite actions for the same state appearing in both trajectories? For example, there are two shortest paths to a goal, one starts with going left and another starts with going right. If you try to generate a state that minimizes the sum of distances to the two states (left and right ones), then you may choose to remain in the middle, which is suboptimal. You wouldn't have this issue with regular IRL techniques, because you can explain both behaviors with future rewards instead of trying to explain every action of the expert using only local state description.

---

> ### Author Response · Authors · 2017-12-21
> **answers**
>
> Thank you very much for your comments.
> We are very happy you understood the benefit of the proposed method.
> And we put the answers to your questions and suspicions, also we updated the paper by following your comments.
>
> >Previous works had used other models for generalizing demonstrations
>
> By our understandings, the other methods are always using the action information of demonstrations, which is simple and straightforward, such as behavior cloning.
> But in this paper, we are tackling without action information for demonstrations.
> If you know, could you please give references about the previous works that are only using observation values?
> If there are similar methods, we want to compare with the proposed method.
>
> And, we are thinking GMMs or GPs could be difficult to predict reward for image inputs.
> It could not adopt the differences between parts of the image, the convolution layer must be needed.
> And we are thinking LSTM and 3D-CNN also consider the time-sequence values, that will be another advantage from these methods.
>
> > - Most IRL methods would work just fine by defining rewards on states only and ignoring actions all together.....
>
> Our assumption is the agent doesn't know the transition function as well as optimal actions.
> We agree if we know the function, we could use this function values to getting expert actions.
>
> > - "We assume that maximizing likelihood of next step prediction in equation 1 will be globally optimized in RL". Could you elaborate more on this assumption? ....
>
> We elaborated the context by your comments.
> "We assume the maximizing likelihood of next step prediction in equation 1 will be globally optimized in RL."
> ->
> "We assume the performing to maximize the likelihood of next step prediction in equation 1 will be leading the maximizing the future reward when the task is deterministic. Because this likelihood is based on similarity with demonstrations which are obtained while an expert agent is performing by maximizing the future reward. Therefore we assume the agent will be maximizing future reward when it takes the action that gets the similar next step to expert demonstration trajectory data \tau. "
>
> > - Related to the previous point: a reward function that makes every step of the expert optimal may not be always exist......
>
> Actually, this is the very important point for this proposed method; we were, of course, thinking this point.
> We thought, if the going this way (terrible states then highly rewarding) is the best way for the RL agent, the expert agent will also take this actions during performing.
> Thus, the proposed method also can find such way.
> However, the samples are not learned by the expert agent, the proposed method cannot find.
> We agree the proposed method is the value function based method.
>
> > - What if in two different trajectories, the expert chooses opposite actions for the same state appearing in both trajectories?....
>
> This is also important; we considered this point.
> If the numbers of multiple trajectories (these future rewards are same by an expert agent) are same, this will have occurred and other IRL techniques also have same problems.
> Because deciding the one way from these multiple trajectories are not possible.
> However, normally (or experimentally), the expert agents that trained RL will take the one choice from multiple trajectories.
> Hence, these points will not be issues of the proposed method.

---

### Decision · Program_Chairs · 2018-01-29
**ICLR 2018 Conference Acceptance Decision**

**Decision:**

Reject

**Comment:**

The paper presents a method for learning from expert state trajectories using a similarity metric in a learned feature space. The approach uses only the states, not the actions of the expert. The reviewers were variously dissatisfied with the novelty, the theoretical presentation, and the robustness of the approach. Though it empirically works better than the baselines (without expert demos) this is not surprising, especially since thousands of expert demonstrations were used. This would have been more impressive with fewer demonstrations, or more novelty in the method, or more evidence of robustness when the agent's state is far from the demonstrations.